# Haprolid Inhibits Tumor Growth of Hepatocellular Carcinoma through Rb/E2F and Akt/mTOR Inhibition

**DOI:** 10.3390/cancers12030615

**Published:** 2020-03-06

**Authors:** Jun Xing, Vikas Bhuria, Khac Cuong Bui, Mai Ly Thi Nguyen, Zexi Hu, Chih-Jen Hsieh, Kathrin Wittstein, Marc Stadler, Ludwig Wilkens, Jun Li, Markus Kalesse, Przemyslaw Bozko, Ruben R. Plentz

**Affiliations:** 1Department of Internal Medicine I, Medical University Hospital, Eberhard Karls Universität Tübingen, 72076 Tubingen, Germany; xingjunjoy@hotmail.com (J.X.); vksbhuria@gmail.com (V.B.); buikhaccuong@gmail.com (K.C.B.); dr.nguyenmaily@gmail.com (M.L.T.N.); chih-jen.hsieh@med.uni-tuebingen.de (C.-J.H.); 2The First Affiliated Hospital of USTC, Division of Life Sciences and Medicine, University of Science and Technology of China, Hefei 230001, China; 3Department of Internal Medicine VIII, Medical University Hospital, 72076 Tübingen, Germany; zexi.hu@med.uni-tuebingen.de; 4Department Microbial Drugs, Helmholtz Centre for Infection Research GmbH (HZI), 38124 Braunschweig, Germany; kathrin.wittstein@helmholtz-hzi.de (K.W.); marc.stadler@helmholtz-hzi.de (M.S.); 5Institute of Pathology, Hannover Regional Hospital, 30043 Hannover, Germany; ludwig.wilkens@krh.eu; 6Institute of Organic Chemistry, Leibniz University of Hannover, 30167 Hannover, Germany; nmlijun@gmail.com (J.L.); markus.kalesse@oci.uni-hannover.de (M.K.); 7Centre of Biomolecular Drug Research, Helmholtz Centre for Infection Research GmbH (HZI), 38124 Braunschweig, Germany; 8Department of Internal Medicine II, Klinikum Bremen Nord, 28755 Bremen, Germany

**Keywords:** HCC, haprolid, Rb/E2F, Akt/mTOR, EMT, cell cycle, apoptosis

## Abstract

Background: Hepatocellular carcinoma (HCC) represents a major health burden with limited curative treatment options. There is a substantial unmet need to develop innovative approaches to impact the progression of advanced HCC. Haprolid is a novel natural component isolated from myxobacteria. Haprolid has been reported as a potent selective cytotoxin against a panel of tumor cells in recent studies including HCC cells. The aims of this study are to evaluate the antitumor effect of haprolid in HCC and to understand its underlying molecular mechanisms. Methods: The efficacy of haprolid was evaluated in human HCC cell lines (Huh-7, Hep3B and HepG2) and xenograft tumors (NMRI-Foxn1^nu^ mice with injection of Hep3B cells). Cytotoxic activity of haprolid was determined by the WST-1 and crystal violet assay. Wound healing, transwell and tumorsphere assays were performed to investigate migration and invasion of HCC cells. Apoptosis and cell-cycle distribution were measured by flow cytometry. The effects of haprolid on the Rb/E2F and Akt/mTOR pathway were examined by immunoblotting and immunohistochemistry. Results: haprolid treatment significantly inhibited cell proliferation, migration and invasion in vitro. The epithelial–mesenchymal transition (EMT) was impaired by haprolid treatment and the expression level of N-cadherin, vimentin and Snail was downregulated. Moreover, growth of HCC cells in vitro was suppressed by inhibition of G1/S transition, and partially by induction of apoptosis. The drug induced downregulation of cell cycle regulatory proteins cyclin A, cyclin B and CDK2 and induced upregulation of p21 and p27. Further evidence showed that these effects of haprolid were associated with Rb/E2F downregulation and Akt/mTOR inhibition. Finally, in vivo nude mice experiments demonstrated significant inhibition of tumor growth upon haprolid treatment. Conclusion: Our results show that haprolid inhibits the growth of HCC through dual inhibition of Rb/E2F and Akt/mTOR pathways. Therefore, haprolid might be considered as a new and promising candidate for the palliative therapy of HCC.

## 1. Introduction

Hepatocellular carcinoma (HCC) is one of the most common cancers globally and its incidence and mortality rate have still been increasing over the past decades [1]. More than half of all HCC patients are diagnosed at advanced stage when curative treatments are not applicable [2]. Currently, palliative first-line treatments with tyrosine kinase inhibitors like sorafenib or lenvantinib are available [3]. Data about immune checkpoint inhibitors are also promising, but new treatment approaches with less side-effects and good response are still needed to overcome this deadly disease [3,4].

Haprolid was isolated from the myxobacterium *Byssovorax cruenta* in 2000. Haprolid has been reported as an effective cytotoxic compound against many tumor cell lines including HCC cells and its IC50 values are within the nanomolar range [5,6]. In this study, we aimed to assess the efficacy of haprolid in HCC and to elucidate its underlying molecular mechanisms of function, thereby providing a rationale for testing it as a novel anticancer drug in clinical trials. This study is the first report for the application of haprolid in an in vitro and in vivo model of HCC.

Many primary pathways altered in human HCC have been described. The retinoblastoma tumor suppressor (Rb) pathway is frequently disrupted in HCC [7,8]. Classically, Rb functions as a central player in cell cycle regulation [9,10]. Mitogenic signals are conveyed through cyclin-dependent kinases (CDKs) which subsequently regulate Rb phosphorylation. Phosphorylated Rb disassociates with E2F transcription factors, a family of cell-cycle-regulated transcription factors which stimulate proliferation, and afterwards initiates the transcription of E2F responsive genes. Rb, p107 and p130 are homologous proteins collectively known as the ‘Rb pocket protein family’ which contributes to the regulation of E2F-responsive genes [11,12]. The key role of the Rb/E2F pathway in regulating cell cycle progression and controlling proliferation offers tremendous potential for the development of therapeutics.

The PI3K/Akt/mTOR signaling cascade is another major signaling pathway implicated in HCC carcinogenesis and plays a central role in driving tumor cell proliferation [8,13]. The PI3K/Akt/mTOR signaling pathway is a prototypal survival pathway which is constitutively activated in a broad range of cancers. This pathway is involved in many types of cellular processes including survival, proliferation, metastasis, metabolism and angiogenesis [13,14,15]. Given the frequent hyperactivation or deregulation of Akt/mTOR pathway in HCC [15,16,17], several small molecule inhibitors targeting this pathway have been developed and are currently undergoing preclinical or clinical trials [8,18].

Our data suggest that haprolid treatment has great antitumor activity in HCC through dual inhibition of Rb/E2F and Akt/mTOR pathways; thus, it may be a new and promising candidate for systemic treatment of HCC.

## 2. Materials and Methods

### 2.1. Cell Culture

Human HCC cell line Huh-7 was obtained from the laboratory of Lars Zender, University Hospital Tübingen, Tübingen, Germany. HepG2 was obtained from the laboratory of Michael Bitzer, University Hospital Tübingen, Germany. Hep3B was purchased from Leibniz Institute DSMZ—German collection of microorganisms and cell cultures. Cell lines were cultured at 37 °C under a 5% CO_2_ environment in DMEM (Thermo Scientific, Darmstadt, Germany) enriched with 10% fetal bovine serum (FBS) (Biochrom, Berlin, Germany) and antibiotics of penicillin/streptomycin (50 units/mL) (Lonza, Verviers, Belgium).

### 2.2. Drug Preparation and In Vitro Treatment

Haprolid was isolated from the culture broth of *Byssovorax cruenta*, using the chromatographic procedure described by Steinmetz et al. [6], and prepared as a 10 mg/mL stock in dimethyl sulfoxide (DMSO) (AppliChem, Darmstadt, Germany). The drug was divided in aliquots, stored at 4 °C and used for in vitro and in vivo experiments. Cells were treated with DMSO as a control or with haprolid in different concentrations (from 0.006 to 6 µg/mL) and were analyzed after 24 to 96 h for certain experiments, namely cell proliferation assay, crystal violet staining, migration assay, invasion assay, tumor spheroid assay, cell cycle analysis and apoptosis assay. The protocols used in this study are described in detail in Appendix A and Methods.

### 2.3. Animals and Treatment

NMRI-Foxn1^nu^ female mice were bought from Charles River Laboratories International (Sulzfeld, Germany). Eight mice were treated intraperitoneally with haprolid (2 mg/kg body weight, on days 1 to 3 each week); six mice were treated with vehicle (DMSO). Treatment was started around 10 weeks of age and mice were sacrificed after 5 weeks of treatment. All treatment solutions were prepared freshly on the day of delivery. Health status of mice was monitored every day. Sizes of visible tumors were measured by caliper and organ tissues were harvested. Tumor volume was calculated using the following formula: Tumor volume V = ((Width)^2^ × (Length)/2). Tumor tissues were fixed in 4% formalin for histology.

### 2.4. Histology and Immunoblotting

Detailed information about antibodies and protocols can be found in the Appendix A and Methods.

### 2.5. Statistics

All the experiments were repeated 2–3 times. The results were analyzed using GraphPad Prism version 7.03 (San Diego, CA, USA) and ImageJ 1.47 (National Institutes of Health, Bethesda, MD, USA) software. The tests included Student’s *t*-test (paired and unpaired) and two-way ANOVA. Differences were considered as statistically significant when the *p*-value was <0.05 (*), <0.01 (**), <0.001 (***), <0.0001 (****) or as not significant (ns).

### 2.6. Study Approval

Mice used in this study were maintained in the animal care of University Hospital Tübingen, Tübingen, Germany. All experimental protocols were reviewed and approved by institutional guidelines for animal care of University Hospital Tübingen (protocol No. M7/17), and all studies were performed according to the methods approved in the protocol.

## 3. Results

### 3.1. Haprolid Treatment Inhibits Proliferation and Metastasis in Human HCC Cells

It was shown before that haprolid (Figure 1A) has antitumor effects against various tumor cells [6]. However, data about HCC and haprolid are limited. Therefore, we determined the cytotoxic effect of haprolid on three human HCC cell lines (Hep 3B, Huh-7 and HepG2) by WST-1 assay. In addition, human fibroblasts, as noncancer cells, were similarly tested to display differential cytotoxicity (Appendix A). The concentrations for 50% of maximal inhibition of cell proliferation (IC50) for Hep3B, Huh-7 and HepG2 cells after 96 h exposure were 0.1533, 0.0313 and 0.0618 µg/mL, respectively (Figure 1B). We continued with concentrations ranging from 0.006 to 6 µg/mL for our further in vitro experiments. Next, we performed cell proliferation and crystal violet assays. As depicted in Figure 1C,D and Appendix A, Haprolid treatment suppressed cell proliferation of Hep3B, Huh-7 and HepG2 cell lines in a dose- and time-dependent manner.

Furthermore, wound healing and Matrigel transwell chamber assays were conducted to examine the effect of haprolid on the cell metastasis. In all cell lines, haprolid treatment induced significant inhibition of wound closure (Figure 2A). Haprolid significantly reduced the number of invasive cells, as illustrated in Figure 2B. Thus, treatment with haprolid can effectively suppress the migration and invasion in human HCC cells. Furthermore, we explored an in vitro three-dimensional tumor spheroid model to mimic physiologic tissue’s microenvironment for investigating drug efficiency of haprolid. As we can see in Figure 3A, these results further confirmed that haprolid treatment significantly inhibited the spheroid growth and invasion in Hep3B and Huh-7 cells. Our experiments showed that incubation of the Huh-7 cell line with haprolid for 10 days suppressed the growth of spheroid size but with no regression, whereas similar incubation of the Hep3B cell line readily resulted in reduction of tumor spheroid size at the concentration of 0.6 µg/mL as early as 4 days. Taken together, these data suggest that haprolid can significantly inhibit proliferation, migration and invasion of human HCC cells. In general, Hep3B cells shows overall stronger inhibitory effect under haprolid treatment compared to Huh-7 and HepG2 cells.

### 3.2. Haprolid Treatment Impairs EMT in HCC Cells

Moreover, based on the findings that haprolid treatment significantly inhibited the HCC cell motility, we continued to investigate the impact of haprolid in EMT. At the molecular level, EMT is frequently determined by the expression level of putative EMT markers: the upregulation of mesenchymal markers, such as cytoskeletal protein vimentin and cell surface protein N-cadherin, and the downregulation of epithelial markers, including the junction protein E-cadherin [19,20]. In order to check whether EMT is impaired after haprolid treatment, we treated HCC cells with increasing dosages of haprolid for 96 h and compared with negative control. In both Hep3B and Huh-7 cells, haprolid treatment resulted in decreased expression of the mesenchymal proteins N-cadherin and vimentin as assessed by Western blot (Figure 3B). However, the expression of epithelial protein E-cadherin was only slightly upregulated (Figure 3B). Thus, we next checked the expression of Snail, a zinc-finger EMT transcription factor that functions as a potent repressor of E-cadherin [21]. As shown in Figure 3B, the level of Snail was markedly diminished after haprolid treatment. These changes indicate that haprolid treatment partially impairs EMT in HCC cells.

### 3.3. Haprolid Inhibits G1/S Transition and Partially Induces Apoptosis of HCC Cells

To figure out the underlying mechanisms for the antiproliferative effect of haprolid, cell cycle and cell apoptosis were analyzed. Fluorescence-activated cell sorting (FACS) assay was carried out to determine the cell cycle profiles upon exposure to different concentrations of haprolid after 48 and 96. Figure 4A shows the distribution of cell cycle profiles after haprolid treatment in all three HCC cell lines. The percentages of the sub-G1-, G1-, S- and G2-M population are listed in Appendix A. In Huh-7 and HepG2 cells, haprolid treatment resulted in a significant accumulation of cells in G1 phase accompanied with a decrease of the cell numbers in S and G2-M phase, which indicated a block of G1/S progression (Figure 4A). Although Hep3B cells also accumulated in the G1 phase at lower concentration (0.06 µg/mL), at higher concentrations of the compound treatment, a large proportion of the cells accumulated in the sub-G1 phase, suggesting an induction of apoptosis (Figure 4A), which was also evidenced by following experiments. Overall, haprolid treatment inhibits the G1/S transition and results in marked cell cycle arrest in all tested cell lines. Moreover, haprolid treatment leads to growth arrest, later on inducing considerable apoptosis in Hep3B cells.

To further substantiate cell cycle analysis results, we next investigated the kinetic expression of cell cycle checkpoint proteins via Western blot (Figure 4B). Consistent with the blocking of G1-to-S phase transition, we found that the expression levels of both cyclin A and cyclin B dynamically decreased. 

Additionally, annexin V and PI staining was performed on live cells to determine whether the cell-cycle-arrest phenotype was associated with the induction of apoptosis. We noticed that exposure to haprolid dose-dependently increased apoptotic cells in Hep3B cell line (Figure 5A,B). Moreover, proteolytic cleavage of poly (ADP-ribose) polymerase (PARP) as an indicative marker of apoptosis was evaluated by quantitative measurement [22]. Immunoblotting revealed distinct PARP cleavage after haprolid treatment in Hep3B cells (Figure 5C,D). Interestingly, Huh-7 and HepG2 cells are not sensitive to the apoptosis-inducing effects of haprolid (Appendix A). These results confirm our previous studies and suggest that Huh-7 and HepG2 cells are relatively resistant to apoptosis and mainly undergo cell cycle arrest under haprolid treatment, while haprolid induces noticeable apoptotic cell death in Hep3B cells following cytostatic effect.

### 3.4. Dual Inhibition of Rb/E2F and Akt/mTOR Pathways by Haprolid

Based on the effect of haprolid on cell cycle, we next examined the expression levels of cell cycle checkpoint proteins and cell cycle modulation factors by Western blot analysis. First, we checked the expression level of Rb, which plays a central role in cell cycle regulation, particularly in initiating DNA replication and division [9]. Figure 6A demonstrated that—aside from the Rb-deficient Hep3B cell line—after haprolid treatment the level of p-Rb expression decreased and Rb changed from the hyperphosphorylated form to the hypophosphorylated form in both Huh-7 and HepG2 cell lines. Moreover, the protein level of E2F-1, the downstream target of Rb, which is the essential transcription factor that regulates cell cycle progression and stimulates proliferation [23], was considerably inhibited after exposure to haprolid in all three HCC cells. Inhibition of Rb phosphorylation and reduction in levels of free E2F-1 appear to play an important role in HCC growth arrest. Haprolid treatment also resulted in a noticeable downregulation of cyclin-dependent kinase 2 (CDK2) in all three HCC cells and upregulation of cyclin-dependent kinase inhibitors (CDKIs) p21 and p27 in both Hep3B and HepG2 cells. We also found upregulation of p21 but not p27 in Huh-7 cells (Figure 6A). Next the phospho-Ser10-histone H3 was checked as a specific marker of mitosis. As expected, in all tested HCC cells, p-histone H3 expression was remarkably downregulated (Figure 6A). Taken together, haprolid induces G1/S phase arrest in HCC cells through contributing to the modulation of cell cycle regulatory proteins and inactivation of the Rb/E2F pathway.

The Akt/mTOR signaling pathway is pivotal to cellular growth and proliferation as well as survival [13]. This pathway is activated in a subgroup of HCC patients and is critical in HCC carcinogenesis [8]. Therefore, we also analyzed the activity of Akt and mTOR. Haprolid treatment significantly decreased the Akt activity at high concentrations: the expression of both total Akt and p-Akt at Ser473, a central regulator of cell survival, were downregulated (Figure 6B). In contrast, low-dose treatment (0.06 µg/mL) with haprolid resulted in weakly increased p-Akt levels in Hep3B and Huh-7 cells, which indicated a dose-dependent differential response to haprolid treatment. Subsequently, decreased p-mTOR levels were detected in Hep3B and Huh-7 cells whereas no detectable changes were observed in HepG2 cells (Figure 6B). Finally, phosphorylation of downstream Akt/mTOR targets, S6K and S6 [24,25], were significantly reduced upon haprolid treatment in all studied cells (Figure 6B). Our results imply that haprolid inhibits cell growth and induces apoptotic cell death in HCC cells through interfering with the Akt/mTOR signaling pathway. Overall, these studies were the first endeavor to reveal the molecular mechanisms of the function of haprolid (Figure 7G).

### 3.5. Haprolid Treatment Inhibits Tumor Growth in NMRI-Foxn1^nu^ Mice

To further evaluate the in vivo antitumor activity of haprolid, we used a subcutaneous xenograft model with Hep3B cells. Hep3B cells were selected, because this cell line was demonstrated to be the most sensitive to haprolid out of the three tested. Mice with tumor xenografts were randomly assigned into two treatment groups and treatment was initiated when xenograft tumors reached a size of ~100 mm^3^. Figure 7A shows a representative image of tumors for control and haprolid (2 mg/kg) groups. As shown in Figure 7A–D, in comparison with the control group, tumor growth in haprolid group was significantly suppressed. In Figure 7D, from 2 weeks of treatment onwards, haprolid induced a statistically significant inhibition of tumor growth (*p* < 0.01) compared to the control group. Over a 5-week period of haprolid treatment, the mean tumor weight was 510.9 ± 84.7 mg compared with 858 ± 75.8 mg in the control group, indicating that the drug reduced tumor growth by 40.4% (Figure 7C). In addition, the total body weight during the entire experiments was not significantly different between both groups, and no side effects were noticed (Appendix A). Correspondingly, mice bearing Hep3B tumors that received haprolid treatment exhibited decreased levels of Ki-67, Akt and p-S6 (Figure 7E,F), as evidenced by IHC analysis. These results, in addition to those obtained in vitro, suggest that haprolid is a potent inhibitor of HCC growth.

## 4. Discussion

Rb/E2F and Akt/mTOR pathways play a pivotal role in the molecular pathogenesis of HCC [8,12,13]. In this study, we report for the first time about the positive antitumor effect of haprolid, a novel natural polyketide–peptide hybrid, in HCC models, both in vitro and in vivo through dual inhibition of Rb/E2F and Akt/mTOR pathways. We found effective inhibition of cell growth in three representative human HCC cell lines: Hep3B, Huh-7 and HepG2. In addition, Hep3B cells demonstrated to be more sensitive to haprolid treatment compared to Huh-7 and HepG2 cells. HDF cells were also tested to demonstrate differential cytotoxicity of haprolid. For further toxicity analysis, primary hepatocytes or liver epithelial cell lines could be used, and more systematically designed experiments should be performed.

EMT plays an important role in the invasive and metastatic abilities of HCC and strongly correlates with tumor progression and the prognosis of patients [19,26,27]. In this study, we demonstrate that haprolid treatment reduces expression of mesenchymal proteins such as N-cadherin, vimentin and Snail in both Hep3B and Huh-7 cells. However, we only detect a subtle upregulation of epithelial marker E-cadherin, arguing for a selective control of EMT by haprolid. Moreover, in vitro wound healing assays, Matrigel chamber assays and 3D tumor spheroid assays provide evidence that the cell motility and invasiveness are remarkably inhibited by haprolid. These findings suggest that haprolid has very good potential for attenuating EMT in HCC and could also be a promising strategy to suppress metastasis of HCC cells.

Relentless cell proliferation and compensatory suppression of apoptosis are “mission critical” events which are required for the development of any and all cancers [28]. Targeting of these pivotal events should be the key to potent and specific anticancer therapies. As it was observed in our study, haprolid treatment could strongly inhibit cell proliferation and negatively regulate the G1/S transition in HCC cells. Cell cycle analysis demonstrated that haprolid significantly decreased cell population in S and G2-M phases. In vitro dynamic analysis revealed that haprolid treatment induced strikingly decreased expression of both cyclin A and cyclin B. Fundamentally, cell-cycle transitions are strictly regulated by cyclins, which bind to and activate their catalytic partners, the CDKs [29,30]. Cyclin A–CDK2 and cyclin A–CDK1 regulate the completion of S phase, whereas cyclin B–CDK1 is essential to take cells into mitosis [29]. In this view, the induction of cell cycle arrest by haprolid is highly correlated with the inhibition of these cyclins. 

In addition, haprolid treatment induced considerable apoptosis on Hep3B cells but not on Huh-7 and HepG2 cells. These findings indicate that induction of apoptosis is highly dependent on the concentration of the drugs as well as the individual susceptibility of the HCC cells. Indeed, extensive literature has disclosed that different HCC cell lines experienced different sensitivity to the induction of apoptosis [31,32,33].

The CDK-Rb-E2F pathway is a critical pathway in cell cycle regulation [34,35]. In the G1 phase, Rb is hypophosphorylated and associates with E2F transcription factors. Transcription is repressed until CDKs phosphorylate Rb and allow unrestrained E2F activity. E2F transcription factors then transcribe genes driving the G1/S cell cycle transition [9,36]. We observed that haprolid treatment effectively inhibited p-Rb and E2F-1 expressions in Huh-7 and HepG2 cells. Interestingly, Rb-deficient Hep3B cells also exhibited significant repression of E2F-1, suggesting that the potential involvement of other pocket proteins p107/p130 compensate for Rb loss, which has been reported by Rivadeneira et al. [37]. We hypothesize that in the Rb-deficient cells, haprolid might function on other pocket proteins p107/p130, although more evidence is needed to further illustrate this. Thus, these observations provided evidence that haprolid induces growth arrest and apoptosis in HCC cells through interfering with the pocket protein/E2F signaling pathway. We also observed that haprolid treatment caused induction of p21 and p27 in Hep3B and HepG2 cells and p21 in Huh-7 cells. One of the reasons that explain the discrepancy between Huh-7 and other tested HCC cells might be due to the point mutation of p53 in Huh-7 cells at codon 220 [38]. p21 and p27, known as the G1-checkpoint CDKIs, coordinate with the internal and external signals which leads to inhibition of cell cycle progression [39]. Loss of expression or function of p21 and p27 has been implicated in the genesis or progression of many human malignancies [40]. Our data show that haprolid suppresses HCC cell growth through the induction of CDKIs and the inhibition of cyclins and CDKs. 

As depicted in the results, we observed that haprolid treatment demonstrated potent inhibition of p-Akt, p-mTOR, p-S6K and p-S6 expressions. Interestingly, we also found that haprolid inhibited Akt phosphorylation at high concentrations but increased signaling at low concentrations. In 2014, Breunig et al. [31] demonstrated that HCC cell lines responded differentially to a group of BRaf and MEK inhibitors depending on the concentration. Low-dose sorafenib treatment of HCC cells increased instead of decreased signaling. Inhibition of the MAPK (mitogen-activated protein kinase) pathway was only achieved by high drug concentrations. Strikingly, we also found the downregulation of full-form Akt. Although the expression level of total Akt is relatively stable in cells, many studies have reported the changes of its level. In 2014, Choi et al. reported that salinomycin, a polyether ionophore antibiotic isolated from *Streptomyces albus*, could reduce total Akt level and sensitize cancer cells to Akt inhibitor MK2206 [41]. In 2008, Mann et al. described how arsenic trioxide, which is a clinical drug to treat acute promyelocytic leukemia, decreased total Akt protein in a caspase-dependent manner [42]. Adachi et al. reported that vascular smooth muscle cells’ growth factors decreased total Akt in a proteasome-dependent way [43]. To further elaborate the mechanism of the downregulation of Akt, we performed experiments by employing proteasome inhibitor MG132 and a pan-caspase inhibitor, Z-VAD. However, the reduction of total Akt level in Hep3B cells after haprolid treatment is neither proteasome nor caspase dependent, as depicted in Appendix A. Furthermore, since Ras/Erk pathway represents another dominant signaling in HCC tumorigenesis, there is a particularly intimate cross-talk between the PI3K/Akt and Ras/Erk pathways, where inhibitors of one pathway will often cause stimulation of the other [44]. In concordance with this consensus, we also determined a remarkable increase of Erk phosphorylation after haprolid treatment, as shown in Appendix A. These findings support the hypothesis that inhibiting only one of the signaling pathways may not be efficacious in HCC treatment, thus promoting the strategy of inhibitor combinations [45]. In summary, our study shows that haprolid suppresses HCC carcinogenesis by interfering with the PI3K/Akt/mTOR signaling pathway. Therefore, this study provides rational support for haprolid’s anticancer effect. 

The in vivo efficacy of haprolid was examined using Hep3B cells grown as xenografts in nude mice. The observation that treatment of these tumor-bearing mice with haprolid substantially impaired tumor development over a 5-week period without detectable side effects is both novel and of potential clinical relevance. Additionally, the IHC staining of tumor tissue sections demonstrates a lower expression of Ki-67, Akt and p-S6 after haprolid treatment. These results are in accordance with our in vitro findings and further confirm the inhibition effect of haprolid on proliferation and Akt signaling. Here we only used the Hep3B xenograft mice model; other appropriate in vivo models including genetically engineered mice models could be used for future studies to validate the results.

## 5. Conclusions

In conclusion, our results indicate that haprolid treatment leads to a strong inhibitory effect of cell growth, migration and invasion in human HCC cell lines. This effect is linked with dual inhibition of Rb/E2F and Akt/mTOR pathways as well as repression of EMT. Cell cycle arrest and apoptosis also contribute to the growth-inhibitory effect of haprolid. Furthermore, we show that haprolid also exhibits antitumor activity in vivo by using an HCC xenograft mouse model. Taken together, our observations highlight the possibility that haprolid might be a new and promising candidate for therapy of HCC. Nevertheless, investigation of combination therapies of haprolid and inhibitors of the Raf/MEK/Erk pathway, the tyrosine kinases (VEGFR, PDGFR), as well as combination with immune-checkpoint inhibitors are warranted in order to try to further extend the antitumor effect of haprolid.

## Figures and Tables

**Figure 1 cancers-12-00615-f001:**
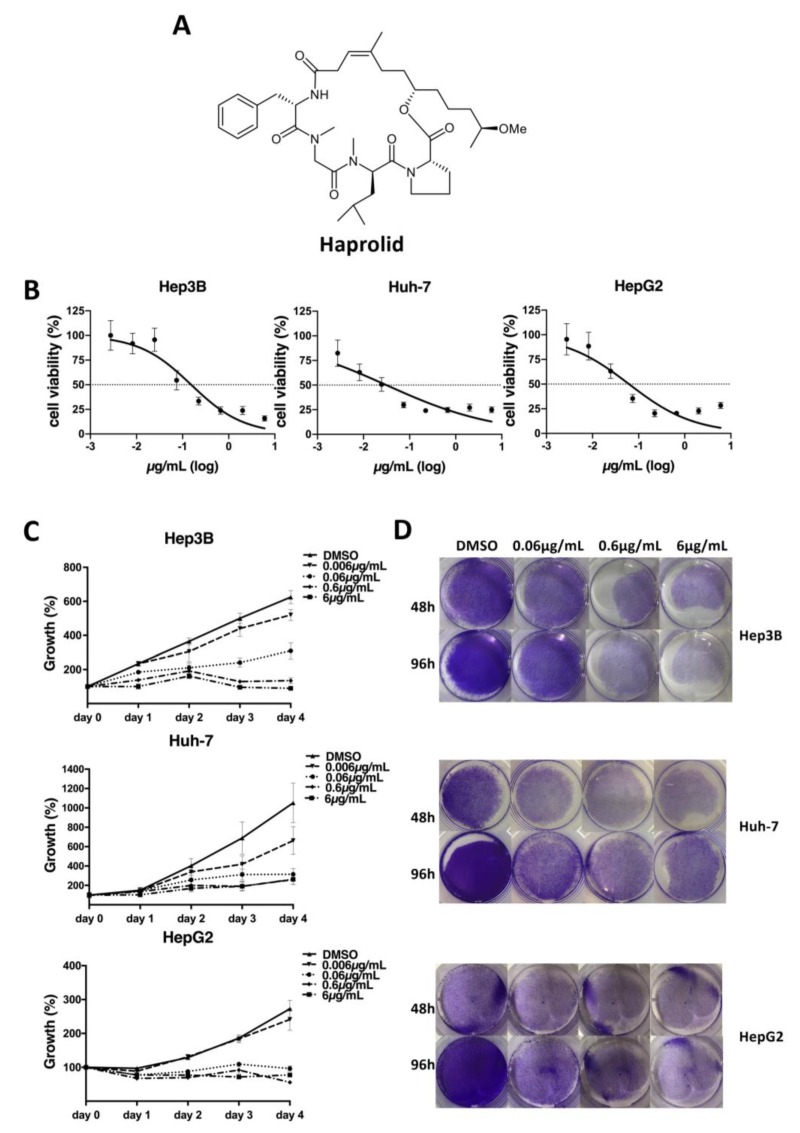
Haprolid treatment inhibits proliferation of human hepatocellular carcinoma (HCC) cells. (**A**) Chemical structure of haprolid. (**B**) Cytotoxic effect of haprolid (IC50): Hep3B, Huh-7 and HepG2 cells were treated for 96 h with increasing concentrations of haprolid (0.001 to 18 µg/mL). WST-1 assay was performed to analyze cellular viability, DMSO was used as negative control. (**C**) Cell proliferation of HCC cells treated with the indicated concentrations of haprolid or DMSO measured by WST-1 assay. The relative cell number was normalized with the control. (**D**) HCC cells were fixed and stained with crystal violet 48 and 96 h after haprolid treatment. Data represent means ± SEM of at least three independent experiments. * *p* < 0.05, ** *p* < 0.01, *** *p* < 0.001, **** *p* < 0.0001.

**Figure 2 cancers-12-00615-f002:**
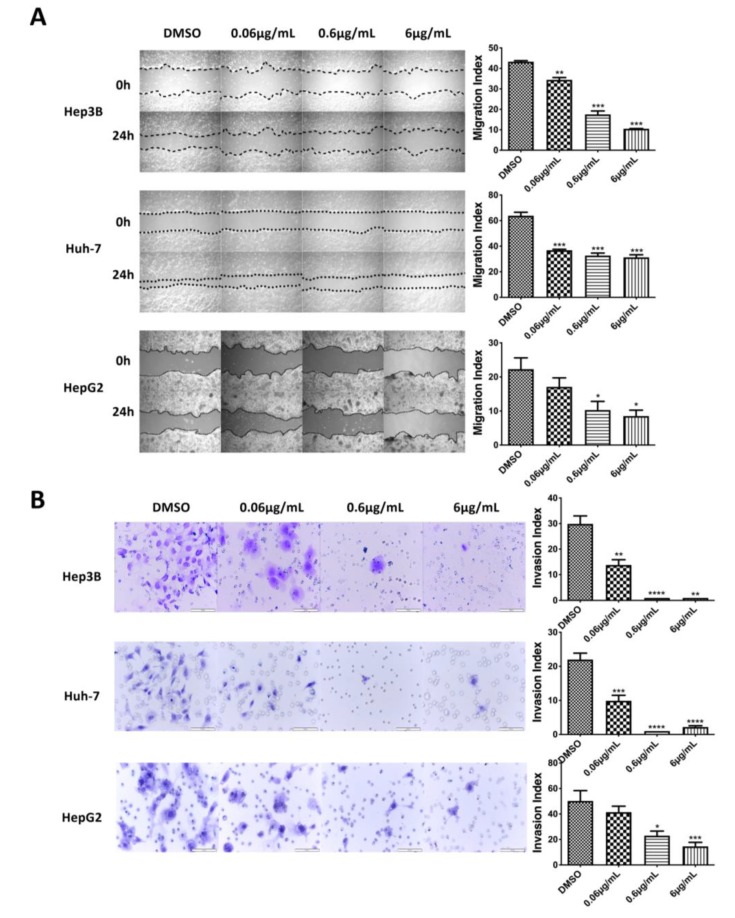
Inhibition of migration and invasion in haprolid-treated HCC cells. (**A**) Wound healing assay was performed for Hep3B, Huh-7 and HepG2 cells after haprolid (0.06, 0.6 and 6 µg/mL) treatment. The dotted lines represent edges of the wound. Photographs were taken at 0 and 24 h under light microscope (40× magnification). The migration index was calculated as described in Section 2 and plotted in bar graphs. (**B**) HCC cells were seeded in Matrigel transwell invasion chambers and treated with indicated concentrations of haprolid for 48 h to investigate the effect on invasiveness. The number of cells that invaded through the membrane was determined under light microscope (200× magnification). Scale bars: 100 µm. Invasion index was calculated as described in Section 2 and plotted in bar graphs. Data represent means ± SEM of at least three independent experiments. * *p* < 0.05, ** *p* < 0.01, *** *p* < 0.001, **** *p* < 0.0001.

**Figure 3 cancers-12-00615-f003:**
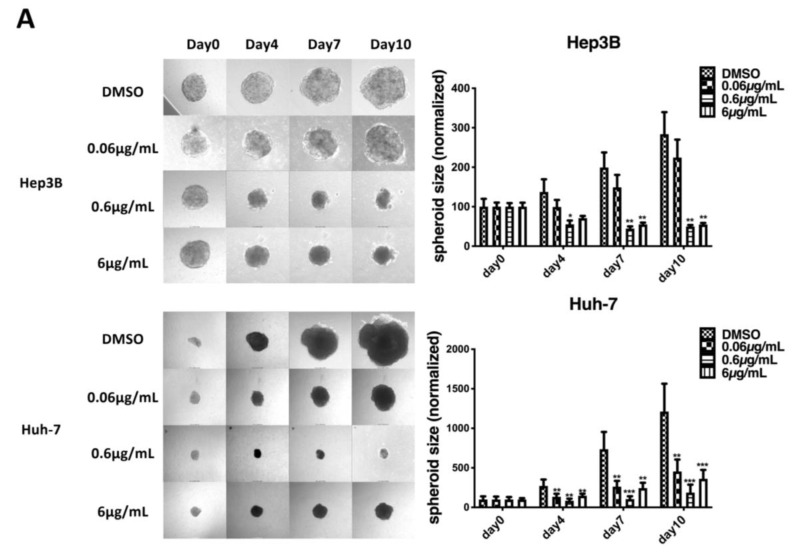
Haprolid treatment inhibits tumor spheroids growth and invasion and partially impairs epithelial–mesenchymal transition (EMT) in human HCC cells. (**A**) Three-dimensional tumor spheroid assay was performed to evaluate effects of haprolid on HCC cell proliferative and invasive ability in extracellular-matrix-like environment. The representative images of spheroids are shown. The changes of spheroids size were monitored and quantified up to 10 days and plotted in bar graphs. (**B**) Hep3B and Huh-7 cells were treated with DMSO or haprolid (0.06, 0.6 and 6 µg/mL) for 96 h. The expressions of EMT markers (N-cadherin, E-cadherin, vimentin and Snail) were analyzed by immunoblotting. Western blotting bands were quantified by Image J and normalized to their respective β-actin then compared with vehicle-treated controls. Data represent means ± SEM of at least three independent experiments. * *p* < 0.05, ** *p* < 0.01, *** *p* < 0.001, **** *p* < 0.0001.

**Figure 4 cancers-12-00615-f004:**
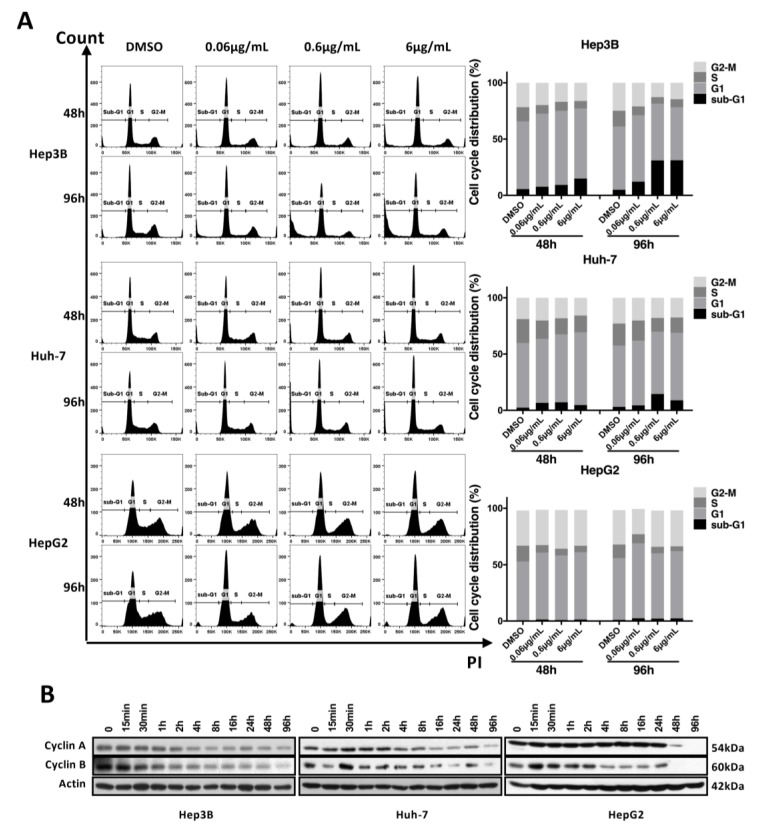
Haprolid treatment inhibits G1/S transition and induces cell cycle arrest in HCC cells. (**A**) HCC cells were treated with vehicle (DMSO) or increasing concentrations of haprolid for 48 and 96 h, fixed and stained with propidium iodide (PI) and subjected to flow cytometric analysis. Representative cell cycle histograms are presented (left panel). Cell cycle distribution in various phases is determined and shown (right panel). (**B**) Cyclin A and B expression levels were measured by Western blotting. Images representative of at least three independent experiments are shown.

**Figure 5 cancers-12-00615-f005:**
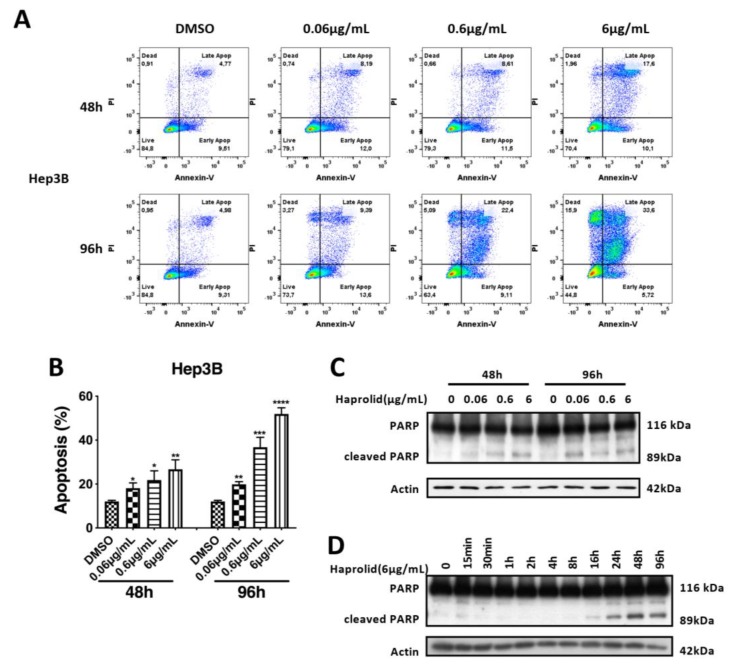
Haprolid treatment induces considerable apoptosis in Hep3B cells. (**A**) Hep3B cells were treated with DMSO or haprolid (0.06, 0.6 and 6 µg/mL) for 48 and 96 h, and the level of apoptosis was measured by staining with annexin V and PI using flow cytometry. Representative FACS measurements are presented. (**B**) The apoptotic positive cells were calculated and plotted in bar graph. (**C**) Expression of PARP and cleaved PARP (a marker for apoptosis) was measured by Western blotting. (**D**) Kinetic analysis of PARP cleavage was performed by Western blotting. Data represent means ± SEM of at least three independent experiments. * *p* < 0.05, ** *p* < 0.01, *** *p* < 0.001, **** *p* < 0.0001.

**Figure 6 cancers-12-00615-f006:**
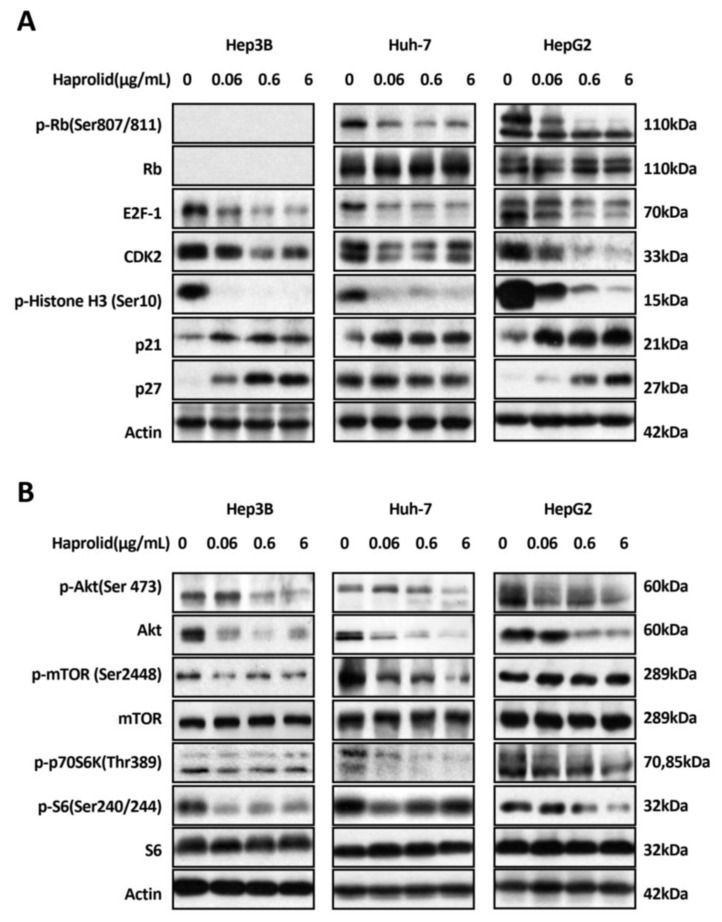
Dual inhibition of Rb/E2F and Akt/mTOR pathways by haprolid. (**A** and **B**) Hep3B, Huh-7 and HepG2 cells were treated with increasing concentrations of haprolid or DMSO for 96 h. Cell lysates were analyzed by Western blotting with antibodies against indicated proteins. β-actin is shown as a loading control. Images representative of at least three independent experiments are shown.

**Figure 7 cancers-12-00615-f007:**
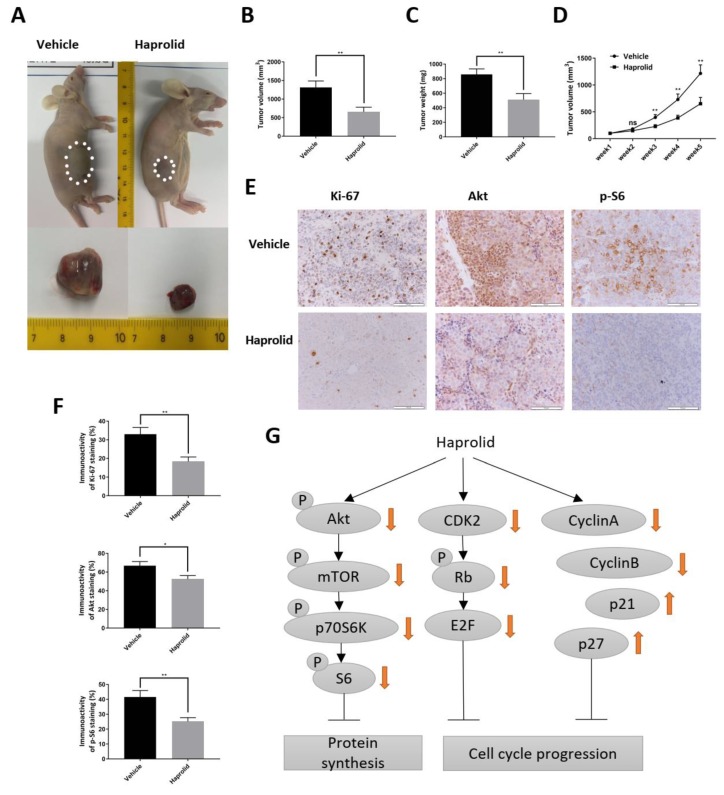
Effect of haprolid on mice bearing Hep3B xenograft tumors. (**A**) Representative image of tumors in control and haprolid-treated (2 mg/kg body weight) mice. Dotted lines sketch out tumors in the flanks. (**B** and **C**) Mean tumor volume and weight in control and haprolid-treated mouse groups (*n* ≥ 5 per group). (**D**) Tumor growth curves. (**E**) Tumor tissue sections were stained with Ki67, Akt and p-S6. Representative IHC staining images were taken at 200× magnification. Scale bars: 100 µm. (**F**) Immunostaining activity of Ki67, Akt and p-S6 was scored and plotted in bar graph. (**G**) The schematic diagram showing the function and mechanism of haprolid in HCC. Data represent means ± SEM of at least three independent experiments. * *p* < 0.05, ** *p* < 0.01, *** *p* < 0.001, **** *p* < 0.0001.

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
