# Peer review of "Haprolid Inhibits Tumor Growth of Hepatocellular Carcinoma through Rb/E2F and Akt/mTOR Inhibition"

_cancers, 2020, doi:10.3390/cancers12030615_

Round 1

Reviewer 1 Report

In this manuscript, Xing et al. study the effect of Haprolid (a natural product isolated from myxobacteria) on HCC cell lines. The authors showed that Haploid treatment inhibited cell proliferation, migration and invasion in part due to Rb/E2F and Akt/mTOR inhibition. Overall, this is a straight forward study and the first study that evaluate the effects of Haprolid in HCC cells. I have a few comments below:

Major comments:

1) Instead of using human dermal fibroblast cells to test for the toxicity of Haploid. A more appropriate cell type/ line would be primary hepatocytes or liver epithelial cell lines such as THLE 2 or 3.

2) EMT is driven by SNAIL, ZEB and basic helix–loop–helix (bHLH) transcription factors that repress epithelial marker genes and activate genes associated with the mesenchymal signature. Did the authors check for TWIST 1 and 2 and ZEB 1 and 2 factors ? These are key EMT factors.

3) Can the authors explain why the DMSO control wells for HepG2 in Fig 1D display similar staining results compared to its corresponding treated counterparts? There seems to be no dose dependent effect on HepG2.

4) Did the authors assess whether Haprolid causes nephrotoxicity in the mice treatment groups besides body weight assessment? In addition, was any upregulation of liver injury markers (e.g. AST and ALT) observed during or at the end of the treatment period?

5) Why did the author choose Hep3B as the in vivo model when it is Rb-deficient? Did the author assess the effects of Haprolid treatment in Huh7 or Hep G2 cancer lines and assess the tissue sections with Akt, p-S6, and p-Rb status? This will give a more complete overall study.

Minor comments:

Please include scale bars for Fig 2A and 3A.

Can the authors justify or discuss how using Haprolid can be used as a palliative therapy as stated in the conclusion?

Reviewer 2 Report

The present in vitro and in vivo study was performed in order to investigate molecular mechanism by which Haprolid inhibit the growth of hepatocellular carcinoma (HCC). Authors reported that Haprolid inhibited cell proliferation, migration and invasion in vitro, being mostly effective in Hep3B cell line, possibly due to induction of apoptosis. Furthermore, it inhibited CDK-Rb-E2F in HepG2 and Huh7 cells. Haprolid at high concentration of 6 mg/ml inhibited but at low concentration induced p-Akt expression, while p-mTOR was decreased in a dose-dependent manner in Hep3B and Huh7, but elevated in HepG2 cells being indicative of mTOR suppression in Hep3B and Huh7 cells.  Akt protein expression was inhibited in all cell lines.

Authors suggested that investigation of combination therapies of Haprolid and inhibitors of the RAF/MEK/ERK pathway, the tyrosine kinases (VEGFR, PDGFR), as well as combination with immune-checkpoint inhibitors are warranted in order to try to further extend the anti-tumor effect of Haprolid.

Major comments

The paper is written in quite good English, numerous experiments were performed, and the data are of interest. It is necessary to explain the choice of the cell lines in the present experiment and their distinct differences in more details. Why Hep3B cell line was the most sensitive? This information appears to be very important. It is necessary to show whether the inhibition of cells growth was statistically significant. I was not able to find the P values in Figures 1 and 4. In Fig.1 P values are shown in figure legend, but asterisks are missing in the Figure. In Fig.3, I was not able to find P value with 1 and 4 asterisks, but they are mentioned in the Figure legend. Similarly, in Fig7, there are no P values with 3 and 4 asterisks, but they are mentioned in the Figure legend. There are some discrepancies in the data obtained in vitro studies. The growth of all liver cancer cell lines (including HepG2) was inhibited, but apoptosis was significantly induced by Haprolid only in Hep3B cells, while p-mTOR was even elevated in HepG2. Does it mean that suppression of Rb has stronger effect in inhibition of the growth in HepG2 cell line, than the effect on mTOR? Or does it mean that Hep3B cell line is much less aggressive? No dose response was observed in vitro study concerning the p-Akt. Authors showed some other studies with differential effect of different doses of the drug on HCC cell lines, trying to explain the observed discrepancies in the Discussion section, however, still more clear explanation is desired. If known, please, give the characteristics of Haprolid and its action at different concentrations, which was observed in previous studies. Detailed characteristics of Haprolid is necessary to understand its action. The statement how the animals were treated, anaesthetized, body and organ weights, food and water consumption measured must be added. It is necessary to mention whether the study was performed accordingly to the Guidelines of National Institute of Health and Public Health Service Policy on the Humane Use and Care of Laboratory Animals, and approved by the Ethics Committee of the Institutional Animal Care at your affiliation.

Reviewer 3 Report

Well written paper. Experiments and results are clearly described. Cahnces and limitations are clearly stated.

Author Response

Thank you very much for your supportive opinions. Our project will continue. We will in the future try to elucidate more detailed mechanism of this drug. Hope this is an interesting and clinic applicable novel drug.

Round 2

Reviewer 1 Report

To address most of my concerns previously, can the authors include discussion/explanation in their results/ discussion section paying particular attention to the following points:

Please include a short discussion on using an appropriate non-cancer cells as a better control for showing differential cytotoxicity in testing novel compounds targeting liver cancer.

Additionally, please discuss the use of more appropriate in vivo model(s) for future studies to validate the results in a clinically meaningful way.

Author Response

Thank you very much for your suggestions. I already made corresponding corrections in the discussion part.
